# Deliberative Explanations: visualizing network insecurities

**Pei Wang and Nuno Vasconcelos**
Department of Electrical and Computer Engineering
University of California, San Diego
{pew062, nvasconcelos}@ucsd.edu

## Abstract

A new approach to explainable AI, denoted *deliberative explanations,* is proposed. Deliberative explanations are a visualization technique that aims to go beyond the simple visualization of the image regions (or, more generally, input variables) responsible for a network prediction. Instead, they aim to expose the deliberations carried by the network to arrive at that prediction, by uncovering the insecurities of the network about the latter. The explanation consists of a list of insecurities, each composed of 1) an image region (more generally, a set of input variables), and 2) an ambiguity formed by the pair of classes responsible for the network uncertainty about the region. Since insecurity detection requires quantifying the difficulty of network predictions, deliberative explanations combine ideas from the literature on visual explanations and assessment of classification difficulty. More specifically, the proposed implementation combines attributions with respect to both class predictions and a difficulty score. An evaluation protocol that leverages object recognition (CUB200) and scene classification (ADE20K) datasets that combine part and attribute annotations is also introduced to evaluate the accuracy of deliberative explanations. Finally, an experimental evaluation shows that the most accurate explanations are achieved by combining non self-referential difficulty scores and second-order attributions. The resulting insecurities are shown to correlate with regions of attributes shared by different classes. Since these regions are also ambiguous for humans, deliberative explanations are intuitive, suggesting that the deliberative process of modern networks correlates with human reasoning.

## 1 Introduction

While deep learning systems enabled significant advances in computer vision, the black box nature of their predictions causes difficulties to many applications. In general, it is difficult to *trust* a system unable to justify its decisions. This has motivated a large literature in explainable AI [22, 42, 21, 44, 43, 26, 28, 13, 18, 49, 3, 8, 34]. In computer vision, most approaches provide visual explanations, in the form of heatmaps or segments that localize image regions responsible for network predictions [37, 51, 49, 31, 20]. More generally, explanations can be derived from attribution models that identify input variables to which the prediction can be attributed [2, 34, 40, 1]. While insightful, all these explanations fall short of the richness of those produced by humans, which tend to reflect the *deliberative nature* of the inference process.

In general, explanations are most needed for inputs that have some types of ambiguity, such that the prediction could reasonably oscillate between different interpretations. In the limit of highly ambiguous inputs it is even acceptable for different systems (or people) to make conflicting predictions, as long as they provide a convincing justification. A prime example of this is visual illusion, such as that depicted in the left of Figure 1, where different image regions provide support for conflicting image interpretations. In this example, the image could depict a "country scene" or a "face". While

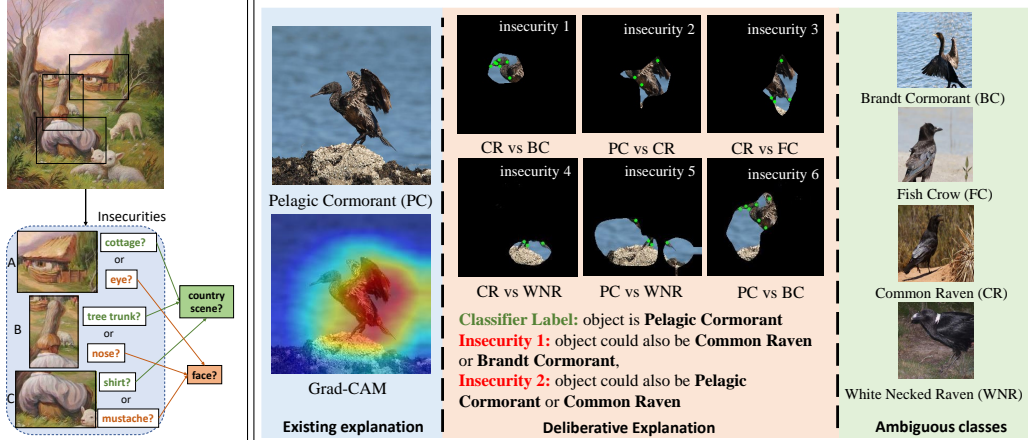

Figure 1: Left: Illustration of the deliberations made by a human to categorize an ambiguous image. Insecurities are image regions of ambiguous interpretation. Right: Deliberative explanations expose this deliberative process. Unlike existing methods, which simply attribute the prediction to image regions (left), they expose the insecurities experienced by the classifier while reaching that prediction (center). Each insecurity consists of an image region and an ambiguity, expressed as a pair of classes to which the region appears to belong to. Examples from the confusing classes are shown in the right. Green dots locate attributes common to the two ambiguous classes, which cause the ambiguity.

one of the explanations could be deemed more credible, both are sensible when accompanied by the proper justification. In fact, most humans would consider the two interpretations while deliberating on their final prediction: "I see a cottage in region A, but region B could be a tree trunk or a nose, and region C looks like a mustache, but could also be a shirt. Since there are sheep in the background, I am going with country scene." More generally, different regions can provide evidence for two or more distinct predictions and there may be a need to deliberate between multiple explanations.

Having access to this deliberative process is important to trust an AI system. For example, in medical diagnosis, a single prediction can appear unintuitive to a doctor, even if accompanied by a heatmap. The doctor's natural reaction would be to ask "what are you thinking about?" Ideally, instead of simply outputting a predicted label and a heat map, the AI system should be able to expose a visualization of its *deliberations,* in the form of a list of image regions (or, more generally, groups of input variables) that support alternative predictions. This list should be ordered by degree of ambiguity, with the regions that generate more uncertainty at the top. This requires the ability to assess the *difficulty* of the decision and the *region of support* of this difficulty in the image.

The quantification of prediction difficulty has received recent interest, with the appearance of several prediction difficulty scoring methods [25, 47, 16, 50, 30, 4]. In this work, we combine ideas from this literature with ideas from the literature on visual explanations to derive a new explanation strategy. Beyond prediction heatmaps, we compute heatmaps of network *insecurities,* listing the regions that support alternative predictions. We refer to these as *deliberative explanations,* since they illustrate the deliberative process of the network. This is illustrated in the right of Figure 1, with an example from the fine-grained CUB birds dataset [48]. On this dataset, where many images contain a single bird, state of the art visualization methods, such as grad-CAM [31] (left inset), frequently produce heatmaps that 1) cover large portions of the bird, and 2) vary little across classes of largest posterior probability, leading to very uninformative explanations. Instead, deliberative explanations provide a list of insecurities (center inset). Each of these consists of 1) an image region and 2) an *ambiguity*, formed by the pair of classes that led the network to be uncertain about the region. Examples of ambiguous classes can also be shown (right inset).

By using deliberative explanations to analyze the decisions of fine-grained deep learning classifiers, we show that the latter perform *intuitive* deliberations. More precisely, we have found that *network insecurities correlate with regions of attributes shared by different classes.* For example, in Figure 1, insecurity 4 is caused by the presence of "black legs" and a "black solid fan-shaped tail," attributes shared by the "Common" and the "White Necked Raven." Similar ambiguities occur for the other insecurities. This observation is quantified by the introduction of a procedure to measure the alignment between network insecurities and attribute ambiguity, for datasets annotated with attributes. We note,

however, that this is not necessary to produce the visualizations themselves, which can be obtained for any dataset. Deliberative visualizations can also leverage most existing visual explanation methods and most methods for difficulty score prediction, as well as benefit from future advances in these areas. Nevertheless, we propose the use of an attribution function of second order, which generalizes most existing visualization methods and is shown to produce more accurate explanations.

We believe that exposing network insecurities will be helpful for many applications. For example, a doctor could choose to ignore the top network prediction, but go with a secondary one after inspection of the insecurities. This is much more efficient than exhaustively analyzing the image for alternatives. In fact, insecurities could help the doctor formulate some hypotheses at the edge of his/her expertise, or help the doctor find a colleague more versed on these hypotheses. Designers of deep learning systems could also gain more insight on the errors of these systems, which could be leveraged to collect more diverse datasets. Rather than just more images, they could focus on collecting the images most likely to improve system performance. In the context of machine teaching [17, 38, 23], deliberative visualizations could be used to enhance the automated teaching of tasks, e.g. labeling of fine-grained image classes, to humans that lack expertise, e.g. Turk annotators. Finally, in cases where deep learning systems outperform humans, e.g. AlphaGo [35], they could even inspire new strategies for thinking about difficult problems, which could improve human performance.

## 2   Related work

**Visualization:** Several works have proposed visualizations of the inner workings of neural networks. Some of these aimed for better understanding of the general computations of the model, namely the semantics of different network units [49, 3, 8]. This has shown that early layers tend to capture low-level features, such as edges or texture, while units of deeper layers are more sensitive to objects or scenes [49]. Network dissection [3] has also shown that modern networks tend to disentangle visual concepts, even though this is not needed for discrimination. However, there is also evidence that concept selectivity is accomplished by a distributed code, based on small numbers of units [8].

Other methods aim to explain predictions made for individual images. One possibility is to introduce an additional network that explains the predictions of the target network. For example, vision-language models can be trained to output a natural language description of the visual predictions. [13] used an LSTM model and a discriminative loss to encourage the synthesized sentences to include class-specific attributes and [18] trained a set of auxiliary networks to produce explanations based on complementary image descriptions. However, the predominant explanation strategy is to compute the contribution (also called importance, relevance, or attribution) of input variables (pixels for vision models) to the prediction, in a post-hoc manner. Perturbation-based methods [51, 49] remove image pixels or segments and measure how this affects the prediction. These methods are computationally heavy and it is usually impossible to test all candidate segments. This problem is overcome by backpropagation-based that produce a 'saliency map' by computing the gradient of the prediction with respect to the input [37]. Variants include backpropagating layer-wise relevances [2], considering a reference input [34], integrated gradients [40] and other modifications of detail [1]. However, prediction gradients have been shown not to be very discriminant, originating similar heatmaps for different predictions [31]. This problem is ameliorated by the CAM family of methods [52, 31], which produce a heatmap based on the activations of the last convolutional network layer, weighting each activation by the gradient of the prediction with respect to it.

In our experience, while superior to simple backpropagation, these methods still produce relatively uninformative heatmaps for many images (e.g. Figure 1). We show that improved visualizations can be obtained with deliberative explanations. In any case, our goal is not to propose a new attribution function, but to introduce a new explanation strategy, deliberative explanations, that visualizes network insecurities about the prediction. This strategy can be combined with any of the visualization approaches above, but also requires an attribution function for the prediction difficulty.

The deliberative explanation seems closely related to counterfactual explanations [14, 45, 10] (or contrastive explanations in some literature [6, 27]), but the two approaches have different motivations. Counterfactual explanations seek regions or language descriptions explaining why an image does not belong to a counter-class. Deliberative explanations seek insecurities, i.e. the regions that make it difficulty for the model to reach its prediction. To produce an explanation, counterfactual methods

only need to consider two pre-specified classes (predicted and counter), deliberative explanations must consider all classes and determine the ambiguous pair for each region.

**Difficulty scores:** Several scores have been proposed to measure the difficulty of a prediction. The most popular is the confidence score, the estimate of the posterior probability of the predicted class produced by the model itself. Low confidence scores identify high probability of failure. However, this score is known to be unreliable, e.g. many adversarial examples [41, 9] are misclassified with high confidence. Proposals to mitigate this problem include the use of the posterior entropy or its maximum and sub-maximum probability values [47]. These methods also have known shortcomings [32], which confidence score calibration approaches aim to solve [5, 25]. Alternatives to confidence scores include Bayesian neural networks [24, 11, 15], which provide estimates of model uncertainty by placing a prior distribution on network weights. However, they tend to be computationally heavy, for both learning and inference. A more efficient alternative is to train an auxiliary network to predict difficulty scores. A popular solution is a failure predictor trained on mistakes of the target network, which outputs failure scores for test samples in a post-hoc manner [50, 30, 4]. It is also possible to train a difficulty predictor jointly with the target network. This is denoted a hardness predictor in [46]. Deliberative explanations can leverage any of these difficulty prediction methods.

## 3 Deliberative explanations

In this section, we discuss the implementation of deliberative explanations. Consider the problem of $C$-class recognition, where an image drawn from random variable $\mathbf{X} \in \mathcal{X}$ has a class label drawn from random variable $Y \in \{1, \ldots, C\}$. We assume a training set $\mathcal{D}$ of $N$ i.i.d. samples $\mathcal{D} = \{(\mathbf{x}_i, y_i)\}_{i=1}^N$, where $y_i$ is the label of image $\mathbf{x}_i$, and a test set $\mathcal{T} = \{(\mathbf{x}_j, y_j)\}_{j=1}^M$. Test set labels are only used to evaluate performance. The goal is to explain the class label prediction $\hat{y}$ produced by a classifier $\mathcal{F} : \mathcal{X} \to \{1, \ldots, C\}$ of the form $\mathcal{F}(\mathbf{x}) = \arg\max_y f_y(\mathbf{x})$, where $\mathbf{f}(\mathbf{x}) : \mathcal{X} \to [0, 1]^C$ is a C-dimensional probability distribution with $\sum_{y=1}^C f_y(\mathbf{x}) = 1$ and is implemented with a convolutional neural network (CNN). The explanation is based on the analysis of a tensor of activations $\mathbf{A} \in \mathbb{R}^{W \times H \times D}$ of spatial dimensions $W \times H$ and $D$ channels, extracted at any layer of the network. We assume that either the classifier or an auxiliary predictor also produce a *difficulty score* $s(\mathbf{x}) \in [0, 1]$ for the prediction. This score is *self-referential* if generated by the classifier itself and *not self-referential* if generated by a separate network.

Following the common practice in the field, explanations are provided as visualizations, in the form of image segments [29, 3, 51]. For deliberative explanations, these segments expose the network insecurities about the prediction $\hat{y}$. An insecurity is a triplet $(\mathbf{r}, a, b)$, where $\mathbf{r}$ is a segmentation mask and $(a, b)$ an ambiguity. This is a pair of class labels such that the network is insecure as to whether the image region defined by $\mathbf{r}$ should be attributed to class $a$ or $b$. Note that none of $a$ or $b$ has to be the prediction $\hat{y}$ made by the network for the whole image, although this could happen for one of them. In Figure 1, $\hat{y}$ is the label "Pelagic Cormorant," and appears in insecurities 2, 5, and 6, but not on the remaining. This reflects the fact that certain parts of the bird could actually be shared by many classes. The explanation consists of a set of $Q$ insecurities $\mathcal{I} = \{(\mathbf{r}_q, a_q, b_q)\}_{q=1}^Q$.

### 3.1 Generation of insecurities

Insecurities are generated by combining attribution maps for both class predictions and the difficulty score $s$. Given the feature activations $\mathbf{a}_{i,j}$ at image location $(i, j)$, the attribution map $m_{i,j}^p$ for prediction $p$ is a map of the importance of $\mathbf{a}_{i,j}$ to the prediction. Locations of activations irrelevant for the prediction receive zero attribution, locations of very informative activations receive maximal attribution. For deliberative explanations, $\mathtt{C} + 1$ attribution maps are computed: a class prediction attribution map $m_{i,j}^c$ for each of the $c \in \{1, \ldots, \mathtt{C}\}$ classes and the difficulty score attribution map $m_{i,j}^s$. Given these maps, the $K$ classes of largest attribution are identified at each location. This corresponds to sorting the attributions such that $m_{i,j}^{c_1} \geq m_{i,j}^{c_2} \geq \ldots \geq m_{i,j}^{c_C}$ and selecting the $K$ largest values. The resulting set $\mathcal{C}(i, j) = \{c_1, c_2, \ldots, c_K\}$ is the *set of candidate classes for location* $i, j$. A set of *candidate class ambiguities* is then computed by finding all class pairs that appear jointly in at least one candidate class list, i.e. $\mathcal{A} = \bigcup_{i,j}\{(a, b)|a, b \in \mathcal{C}(i, j), a \neq b\}$, and an *ambiguity map*

$$m_{i,j}^{(a,b)} = f(m_{i,j}^a, m_{i,j}^b, m_{i,j}^s) \tag{1}$$

is computed for each ambiguity in $\mathcal{A}$. While currently $f(.)$ consists of the product of its arguments, we will investigate other possibilities in the future. The goal is for $m_{i,j}^{(a,b)}$ to be large only when location $(i,j)$ is deemed difficult to classify (large difficulty attribution $m_{i,j}^s$) *and* this difficulty is due to large attributions to *both* classes $a$ and $b$. Finally, the ambiguity map is thresholded to obtain the segmentation mask $\mathbf{r}(a,b) = \mathbb{1}_{m_{i,j}^{(a,b)} > T}$, where $\mathbb{1}_{\mathcal{S}}$ is the indicator function of set $\mathcal{S}$ and $T$ a threshold. The ambiguity $(a,b)$ and the mask $\mathbf{r}(a,b)$ form an *insecurity*.

## 3.2 Attribution maps

Attribution map $m_{i,j}^p$ is a measure of how the activations $\mathbf{a}_{i,j}$ at location $(i,j)$ contribute to prediction $p$. This could be a class prediction or a difficulty prediction. In this section, we make no difference between the two, simply denoting $p = g_p(\mathbf{A})$, where $\mathbf{g}$ is the mapping from activation tensor $\mathbf{A}$ into prediction vector $\mathbf{g}(\mathbf{A}) \in [0,1]^P$. For class predictions $P = C$, the prediction is a class, and $g_y(\mathbf{A}(\mathbf{x})) = f_y(\mathbf{x})$. For difficulty predictions $P = 1$, the prediction is a difficulty score, and $g(\mathbf{A}(\mathbf{x})) = s(\mathbf{x})$. Apart from popular $1^{st}$ order attribution maps [33, 31, 40], we also attempt the attribution map based on a second-order Taylor series expansion of $g_p$ at each location $(i,j)$ and some approximations that are discussed in Appendix. This has the form

$$m_{i,j}^p = [\nabla g_p(\mathbf{A})]_{i,j}^T \mathbf{a}_{i,j} + \frac{1}{2} \mathbf{a}_{i,j}^T [\mathbf{H}(\mathbf{A})]_{i,j} \mathbf{a}_{i,j}, \tag{2}$$

where $\mathbf{H}(\mathbf{A}) = \nabla^2 g_p(\mathbf{A})$ is the Hessian matrix of $g_p$ at $\mathbf{A}$. In Appendix, we also show that most attribution maps previously used in the literature are special cases of (2), based on a first order Taylor expansion. In section 5.2, we show that the second order approximation leads to more accurate results.

## 3.3 Difficulty scores

For class attributions, $g_p(\mathbf{A}) = f_p(\mathbf{x})$, i.e. the $p^{th}$ output of the softmax at the top of the CNN. For difficulty scores, $g(\mathbf{A})$ is the output a single unit that produces the score. The exact form of the mapping depends on the definition of the latter. We consider three scores previously used in the literature. The *hesitancy score* is defined as the complement of the largest class posterior probability in the work, i.e.

$$s^{he}(\mathbf{x}) = 1 - \max_y f_y(\mathbf{x}). \tag{3}$$

This can be implemented by adding a max pooling layer to the softmax outputs. The score is large when the confidence of the classification prediction is low. The *entropy score* [47] is the normalized entropy of the softmax probability distribution defined by

$$s^e(\mathbf{x}) = -\frac{1}{\log C} \sum_y f_y(\mathbf{x}) \log f_y(\mathbf{x}). \tag{4}$$

These two scores are self-referential. The final score, denoted the *hardness score* [46], relies on a classifier-specific hardness predictor $\mathcal{S}$, which is jointly trained with the classifier $\mathcal{F}$. $\mathcal{S}$ thresholds the output of a network $s(\mathbf{x}) : \mathcal{X} \to [0,1]$ whose output is a sigmoid unit. The difficulty score is

$$s^{ha}(\mathbf{x}) = s(\mathbf{x}). \tag{5}$$

# 4 Evaluation of deliberative explanations

Explanations are usually difficult to evaluate, since explanation ground truth is usually not available. While some previous works only show visualizations [40, 39], two major classes of evaluation strategies were used. One possibility is to perform Turk experiments, e.g. measuring whether humans can predict a class label given a visualization, or identify the most trustworthy of two models that make identical predictions from their explanations [31]. In this paper, we attempted to measure whether, given an image and an insecurity produced by the explanation algorithm, humans can predict the associated ambiguities. While this strategy directly measures how intuitive the explanations appear to humans, it requires experiments that are somewhat cumbersome to perform and difficult to replicate. A second evaluation strategy is to rely on a proxy task, such as localization [52, 31]

on datasets with object bounding boxes. This is much easier to implement and replicate and is the approach that we pursue in this work. However, this strategy requires ground-truth for insecurities. For this, we leverage datasets annotated with parts and attributes. More precisely, we equate segments to *parts,* and define insecurities as *ambiguous parts,* e.g., object parts common to multiple object classes or scene parts (e.g. objects) shared between different scene classes. To quantify part ambiguity, parts are annotated with attributes[1]. Specifically, the $k^{th}$ part is annotated with a semantic descriptor of $D_k$ *attribute values.* For example, in a bird dataset, the "eye" part can have color attribute values "green," "blue," "brown," etc. The descriptor is a probability distribution over these attribute values, characterizing the variability of attribute values of the part under each class. The attribute distribution of part $k$ under class $c$ is denoted $\phi_c^k$. The *strength of the ambiguity* between classes $a$ and $b$, according to segment $k$, is then defined as $\alpha_{a,b}^k = \gamma(\phi_a^k, \phi_b^k)$, where $\gamma$ is a similarity measure. This declares as ambiguous parts that have similar attribute distributions under the two classes.

To generate insecurity ground-truth, ambiguity strengths $\alpha_{a,b}^k$ are computed for all parts $\mathbf{p}_k$ and class pairs $(a,b)$. The $M$ insecurities $\mathcal{G} = \{(\mathbf{p}_i, a_i, b_i)\}_{i=1}^M$ of largest ambiguity strength are selected as the insecurity ground-truth. Two metrics are used for evaluation, depending on the nature of part annotations. For datasets where parts labelled with a single location (usually the center of mass of the part), i.e. $\mathbf{p}_i$ is a point, the quality of insecurity $(\mathbf{r}, a, b)$ is computed by precision (P) and recall (R), where $P = \frac{J}{|\{k|\mathbf{p}_k \in \mathbf{r}\}|}$, $R = \frac{J}{|\{i|(\mathbf{p}_i, a_i, b_i) \in \mathcal{G}, a_i = a, b_i = b\}|}$ and $J = |\{i|\mathbf{p}_i \in \mathbf{r}, a_i = a, b_i = b\}|$ is the number of included ground-truth insecurities by the generated insecurity. For datasets where parts have segmentation masks, the quality of $(\mathbf{r}, a, b)$ is computed by the intersection over union (IoU) metric $IoU = \frac{|\mathbf{r} \cap \mathbf{p}|}{|\mathbf{r} \cup \mathbf{p}|}$, where $\mathbf{p}$ is the part in $\mathcal{G}$ with ambiguity $(a, b)$ and largest overlap with $\mathbf{r}$. Curves of precision-recall curves and IoU are generated by varying the threshold $T$ used to transform the ambiguity maps of (1) into insecurity masks $\mathbf{r}(a, b)$. For each image, $T$ is chosen so that insecurities cover from 1% to 90% of the image, with steps of 1%.

## 5 Experiments

In this section we discuss experiments performed to evaluate the quality of deliberative explanations.

### 5.1 Experimental setup

**Dataset:** Experiments were performed on the CUB200 [48] and ADE20K [53] datasets. CUB200 [48] is a dataset of fine-grained bird classes, annotated with parts. 15 part locations (points) are annotated including back, beak, belly, breast, crown, forehead, left/right eye, left/right leg, left/right wing, nape, tail and throat. Attributes are defined per part according to [48] (see Appendix). ADE20K [53] is a scene image dataset with more than 1000 scene categories and segmentation masks for 150 objects. In this case, objects are seen as scene parts and each object has a single attribute, which is its probability of appearance in a scene. Both datasets were subject to standard normalizations. All results are presented on the standard CUB200 test set and the official validation set of ADE20K. Since deliberative explanations are most useful for examples that are difficult to classify, explanations were produced only for the 100 test images having largest difficulty score on each dataset. All experiments used candidate class sets $\mathcal{C}(i, j)$ of 3 members and among top 5 predictions, and were ran three times.

**Network:** Unless otherwise noted, VGG16 [36] is used as the default architecture for all visualizations. This is because it is the most popular architecture in visualization papers. Its performance was also compared to those of ResNet50 [12] and AlexNet [19]. All classifiers and predictors are trained by standard strategies [36, 12, 19, 47, 46]. The widely used last convolutional layer output with positive contributions in the visualization literature [20, 52, 31] is used.

**Evaluation:** On CUB200 all semantic descriptors $\phi_c^k$ are multidimensional, i.e. $D_k > 1, \forall k$. In this case, ambiguity strengths $\alpha_{a,b}^k$ are computed with $\gamma(\phi_a^k, \phi_b^k) = e^{-\{D(\phi_a^k || \phi_b^k) + D(\phi_b^k || \phi_a^k)\}}$ [7] averagely among all attributes, where $D(.||.)$ is the Kullback–Leibler divergence. The number of $M$ of ground-truth insecurities is set to the 20% triplets $(\mathbf{p}_i, a_i, b_i)$ in the dataset of strongest ambiguity. Since parts are labelled with points, insecurity accuracy is measured with precision and recall. On ADE20K, the semantic descriptors $\phi_c^k$ are scalar, i.e. $D_k = 1, \forall k$, and $\phi_c^k$ is the probability of

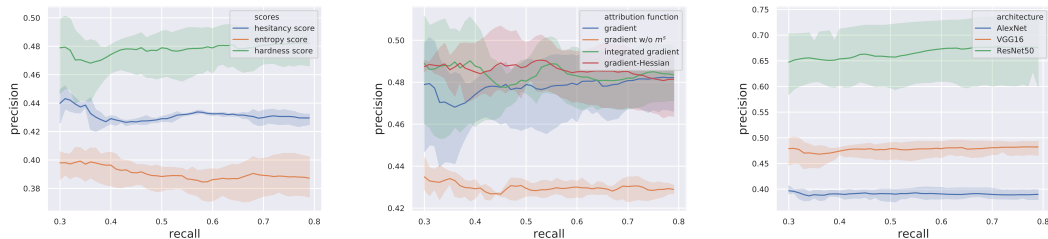

Figure 2: Impact of different algorithm components on precision-recall (CUB200). Left: difficulty scores, center: attribution functions, right: network architectures.

| Methods | 10% | 20% | 30% | 40% | 50% | Avg. |
|---|---|---|---|---|---|---|
| Hesitancy score | 8.32(0.05) | 15.62(0.01) | 22.25(0.02) | 28.45(0.06) | 34.31(0.11) | 21.79(0.03) |
| Entropy score [47] | 8.16(0.06) | 15.10(0.08) | 21.26(0.07) | 26.92(0.18) | 32.23(0.30) | 20.73(0.09) |
| Hardness score [46] | **8.63**(0.12) | **16.59**(0.16) | **24.14**(0.19) | **31.34**(0.22) | **38.29**(0.24) | **23.80**(0.19) |
| Gradient [33] | 8.63(0.12) | 16.59(0.16) | 24.14(0.19) | 31.34(0.22) | 38.29(0.24) | 23.80(0.19) |
| Gradient w/o $m^s$ | 8.54(0.17) | 16.35(0.44) | 23.70(0.77) | 30.67(1.16) | 37.39(1.59) | 23.33(0.82) |
| Int. grad. [40] | 8.70(0.12) | 16.75(0.20) | 24.37(0.27) | 31.60(0.31) | 38.56(0.30) | 23.99(0.24) |
| Gradient-Hessian | **8.86**(0.20) | **17.00**(0.29) | **24.65**(0.32) | **31.92**(0.35) | **38.88**(0.34) | **24.26**(0.30) |
| AlexNet | 8.53(0.16) | 16.03(0.39) | 22.97(0.65) | 29.50(0.90) | 35.71(1.17) | 22.55(0.65) |
| VGG16 | **8.63**(0.12) | **16.59**(0.16) | **24.14**(0.19) | **31.34**(0.22) | **38.29**(0.24) | **23.80**(0.19) |
| ResNet50 | 8.23(0.14) | 15.80(0.21) | 22.92(0.24) | 29.76(0.27) | 36.30(0.26) | 22.60(0.22) |

Table 1: Impact of algorithm components on IoU precision (ADE20K).

occurrence of part (object) $k$ in scenes of class $c$. This is estimated by the relative frequency with which the part appears in scenes of class $c$. Only parts such that $\phi_c^k > 0.3$ are considered. Ambiguity strengths are computed with $\gamma(\phi_a^k, \phi_b^k) = \frac{1}{2}(\phi_a^k + \phi_b^k)$. This is large when object $k$ appears very frequently in both classes, i.e. the object adds ambiguity, and smaller when this is not the case. Due to the sparsity of the matrix of ambiguity strengths $\alpha_{a,b}^k$, the number $M$ of ground-truth insecurities is set to the $1\%$ triplets of strongest ambiguity. Insecurity accuracy is measured with the IoU metric.

## 5.2 Ablation study

**Difficulty Scores:** Figure 2 (left) shows the precision-recall curves obtained on CUB200 for different difficulty scores. The top section of Table 1 presents the corresponding analysis for IoUs on ADE20K. Some conclusions can be drawn. First, all methods substantially outperform random insecurity extraction, whose precision is around $20\%$. Second, precision curves are fairly constant while IoU increases substantially above $30\%$ image coverage. This suggests that insecurities tend to cover ambiguous image regions, but segmentation is imperfect. Third, in both cases, the hardness score substantially outperforms the remaining scores. This suggests that self-referential difficulty scores should be avoided. The hardness score is used in the remaining experiments.

**Attribution Function:** Deliberative explanations are compatible with any attribution function. Figure 2 (center) and the second section of Table 1 compare the $2^{nd}$ order approximation of (2), denoted 'gradient-Hessian,' to the more popular $1^{st}$ order approximation [33] consisting of the first term of (2) only ('gradient'), and the integrated gradient of [40]. 'Gradient' is also implemented without using the difficulty score attribution map in (1), denoted 'gradient w/o $m^s$'. A few conclusions are possible. First, gradient-Hessian always outperforms gradient generally but on ADE20K there is no significant difference. [39] found experimentally that gains of second-order term decrease as the number of classes increases. This could explain why no clear gain on ADE20K ($> 1000$ categories) compared with CUB200 (200 categories). Second, 'gradient w/o $m^s$' has the worst performance of all methods, showing that difficulty attributions are important for deliberative explanations.

**Network Architectures:** Figure 2 (right) and the bottom section of Table 1 compare the explanations produced by ResNet50, VGG16, and AlexNet. Since for the ResNet the second- and higher-order of

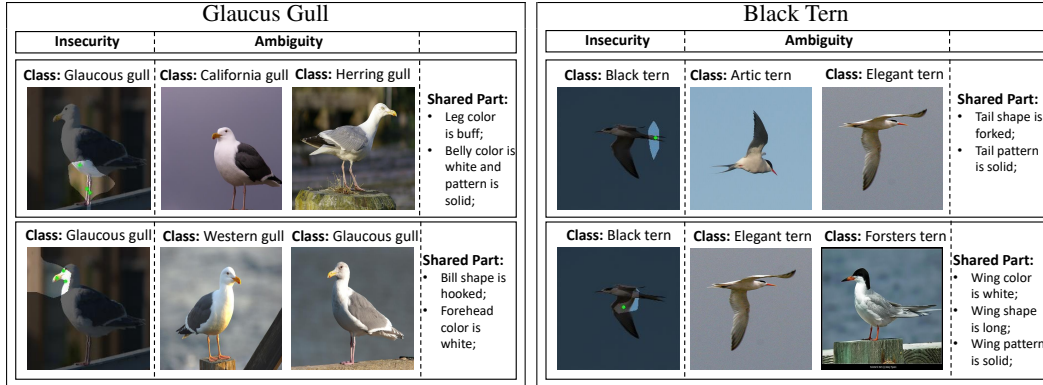

Figure 3: Deliberative visualizations for two images from CUB. Left: a Glaucus Gull creates two insecurities. Top: the insecurity shown on the left elicits ambiguity between the California and Herring Gull classes. The attributes of the shared part are listed on the right. Bottom: insecurity with ambiguity between Western and Glaucous Gull classes. Right: similar for Black Tern. In all insecurities, green dots locate the shared part.

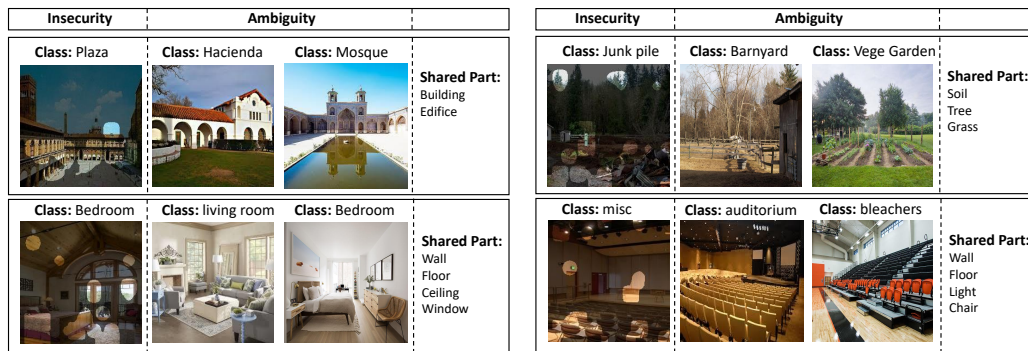

Figure 4: Deliberative visualizations for four images from ADE20K.

(2) are zero (see a proof in [31]), we used the first order approximation on these experiments. On CUB200, AlexNet performed the worst and ResNet50 the best. Interestingly, although ResNet50 and VGG16 have similar classification performance, the ResNet insecurities are much more accurate than those of VGG16. This suggests that the ResNet architecture uses more intuitive, i.e. human-like, deliberations. On ADE20K, the classification task is harder ($< 60\%$ mean accuracy). There is no clear difference among three architectures.

## 5.3 Deliberative explanation examples

We finish by discussing some deliberative visualizations of images from the two datasets. These results were obtained with the hardness score of (5) and gradient-based attributions on ResNet50. Figure 3 shows two examples of two insecurities each. On the left side of the figure, an insecurity on the leg/belly region of a 'Glaucus gull' is due to and ambiguity with classes 'California gull' and 'Herring gull' with whom it shares leg color 'buff', belly color 'white', and belly pattern 'solid'. A second insecurity emerges in the bill/forhead region of the gull, due to an ambiguity between 'Glaucus gull' and 'Western gull' with whom the 'Glaucus gull' shares a 'hooked' bill shape and a 'white' colored forehead. The right side of the figure shows insecurities for a 'Black tern,' due to a tail ambiguity between 'Artic' and 'Elegant' terns and a wing ambiguity between 'Elegant' and 'Forsters' terns. Figure 4 shows single insecurities from four images of ADE20K. In all cases, the insecurities correlate with regions of attributes shared by different classes. This shows that deliberative explanations unveil truly ambiguous image regions, generating intuitive insecurities that help understand network predictions. Note, for example, how the visualization of insecurities tends to highlight classes that are semantically very close, such as the different families of gulls or terns and class subsets such as 'plaza', 'hacienda', and 'mosque' or 'bedroom' and 'living room'. All of this suggests that the deliberative process of the network correlates well with human reasoning.

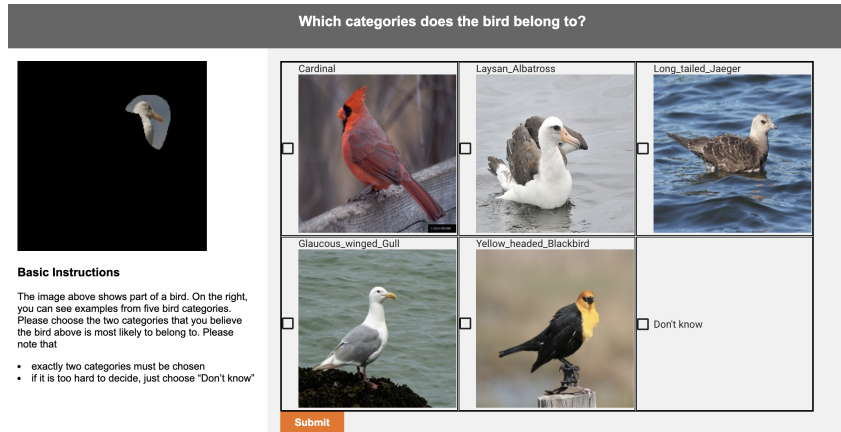

Figure 5: MTurk interface

# 6 Human evaluation results

The designed interface of the human experiment with an example is given in Figure 5. The region of support of the uncertainty is shown on the left and examples from five classes are displayed on the right. These include the two ambiguous classes found by the explanation algorithm, the "Laysan Albatross" and the "Glaucous Winged Gull". If the Tuker selects these two classes there is evidence that the insecurity is intuitive. Otherwise, there is evidence that it is not.

We performed a preliminary human evaluation for the generated insecurities on MTurk. Given an insecurity $(\mathbf{r}, a, b)$ found by the explanation algorithm, turkers were shown $\mathbf{r}$ and asked to identify $(a, b)$ among 5 classes (for which a random image was displayed) including the two classes $a$ and $b$ found by the algorithm. As a comparison, randomly cropped regions with the same size as insecurities were also shown to turkers. We found that turkers agreed amongst themselves on $a$ and $b$ for $59.4\%$ of the insecurities and $33.7\%$ of randomly cropped regions. Turkers agreed with the algorithm for $51.9\%$ of the insecurities and $26.3\%$ of the random crops. This shows that 1) insecurities are much more predictive of the ambiguities sensed by humans, and 2) the algorithm predicts those ambiguities with exciting levels of consistency, given the very limited amount of optimization of algorithm components that we have performed so far. In both cases, the "Don't know" rate was around $12\%$.

# 7 Conclusion

In this work, we have presented a novel explanation strategy, deliberative explanations, aimed at visualizing the deliberative process that leads a network to a certain prediction. A procedure was proposed to generate these explanations, using second order attributions with respect to both classes and a difficulty score. Experimental results have shown that the latter outperform the first-order attributions commonly used in the literature, and that referential difficulty scores should be avoided, whenever possible. The strong annotations are just needed to evaluate explanation performance, i.e. on the test set. Hence, the requirement for annotations is a limitation but only for the evaluation of deliberative explanation methods, not for their use by practitioners. Finally, deliberative explanations were shown to identify insecurities that correlate with human notions of ambiguity, which makes them intuitive.

## Acknowledgement

This work was partially funded by NSF awards IIS-1546305, IIS-1637941, IIS-1924937, and NVIDIA GPU donations.

## Footnotes

[1] It should be noted that part and attribute annotations are only required to evaluate the accuracy of insecurities, not to compute the visualizations. These require no annotation.

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
