[Supplementary Material]

# Supplementary materials for: Deliberative Explanations: visualizing network insecurities

**Pei Wang and Nuno Vasconcelos**
Department of Electrical and Computer Engineering
University of California, San Diego
{pew062, nvasconcelos}@ucsd.edu

## Appendix 1: Attribution Maps

An attribution map can be created by performing a first-order Taylor series expansion of the mapping $g_p$ at each location $(i, j)$. Performing the Taylor expansion in the neighborhood of a reference tensor of feature responses $\mathbf{A}^0 \in \mathbb{R}^{W \times H \times D}$ leads to

$$m_{i,j}^p = g_p(\mathbf{A}^0) + [\nabla g_p(\mathbf{A}^0)]_{i,j}^T[\mathbf{a}_{i,j} - \mathbf{a}_{i,j}^0] + \mathcal{O}^n \tag{1}$$

Several attribution approaches in the literature [8, 3, 5, 2, 4, 6] can be seen as implementing a map of this form, under the assumptions that $\mathbf{A}^0 = \mathbf{0}$ and $g_p$ is a linear mapping that passes through the origin, i.e. $g_p(\mathbf{0}) = \mathbf{0}$ and $\nabla g_p(\mathbf{A}) = \nabla g_p(\mathbf{A}^0)$. In this case,

$$m_{i,j}^p = [\nabla g_p(\mathbf{A})]_{i,j}^T \mathbf{a}_{i,j}. \tag{2}$$

Different approaches use maps of this type, focusing on alternative ways to compute the gradient [1], e.g. using maps of the form

$$m_{i,j}^p = \langle \nabla g_p(\mathbf{A}) \rangle^T \mathbf{a}_{i,j}. \tag{3}$$

where $\langle \nabla g_p(\mathbf{A}) \rangle$ is some gradient average [3, 6]. Rather than this, we consider the second-order Taylor series expansion of $g_p$

$$m_{i,j}^p = g_p(\mathbf{A}^0) + [\nabla g_p(\mathbf{A}^0)]_{i,j}^T[\mathbf{a}_{i,j} - \mathbf{a}_{i,j}^0] + \frac{1}{2}[\mathbf{a}_{i,j} - \mathbf{a}_{i,j}^0]^T[\mathbf{H}(\mathbf{A}^0)]_{i,j}[\mathbf{a}_{i,j} - \mathbf{a}_{i,j}^0] + \mathcal{O}^n \tag{4}$$

where $\mathbf{H}(\mathbf{A}^0) = \nabla^2 g_p(\mathbf{A}^0)$ is the Hessian matrix of $g_p$ at $\mathbf{A}^0$. We then make similar assumptions, namely that $\mathbf{A}^0 = \mathbf{0}$ and $g_p(\mathbf{0}) = \mathbf{0}$. However, rather than assuming that $g_p$ is linear, we assume that it is a second order function and that $\mathbf{A} \approx \mathbf{0}$, from which it follows that $\nabla g_p(\mathbf{A}^0) \approx \nabla g_p(\mathbf{A})$ and $\mathbf{H}(\mathbf{A}^0) \approx \mathbf{H}(\mathbf{A})$. This leads to

$$m_{i,j}^p = [\nabla g_p(\mathbf{A})]_{i,j}^T \mathbf{a}_{i,j} + \frac{1}{2}\mathbf{a}_{i,j}^T[\mathbf{H}(\mathbf{A})]_{i,j}\mathbf{a}_{i,j}. \tag{5}$$

While the use of gradient and Hessian averages, as in (3), could also be used, we have not yet considered such variants.

## Appendix 2: Attribute Assignment

The parts and attributes of the CUB200 dataset [7] are listed in Table 1.

## Appendix 3: More Success Cases

More success case images are shown in Figure 1 for ADE20K and 2 for CUB200.

| Parts | Attributes |
|---|---|
| back | back color, back pattern |
| beak | bill shape, bill length, bill color |
| belly | belly color, belly pattern |
| breast | breast color, breast pattern |
| crown | crown color, forehead color, head pattern |
| forehead | forehead color, head pattern |
| left/right eye | eye color, head pattern |
| left/right leg | leg color |
| left/right wing | wing color, wing shape, wing pattern |
| nape | nape color |
| tail | tail shape, upper tail color, under tail color, tail pattern |
| throat | throat color, head pattern |

Table 1: Attributes assignments on CUB200 [7]

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

| Insecurity | Ambiguity | |
|---|---|---|

**Class:** Nunnery    **Class:** Nunnery    **Class:** Fortress

**Shared Part:**
Building
Edifice
Grass

**Class:** Abbey    **Class:** Mental institution outdoor    **Class:** Guardroom

**Shared Part:**
Sky
Window

**Class:** Mountain    **Class:** Moor    **Class:** Heath

**Shared Part:**
Person

**Class:** Bathhouse    **Class:** Inn outdoor    **Class:** House

**Shared Part:**
Sky
Grass
Plant
Building

**Class:** Questionable    **Class:** Inn indoor    **Class:** Living room

**Shared Part:**
Wall
Floor
Celling
Window
Chair
Lamp

**Class:** Theater outdoor    **Class:** Entrance    **Class:** Bank outdoor

**Shared Part:**
Building
Road
Sidewalk

Figure 1: Success case of deliberative visualizations for images from ADE20K [9].

| Insecurity | Ambiguity | |
|---|---|---|
| **Class:** American crow | **Class:** Common raven    **Class:** American crow | **Shared Part:**<br>• Leg color is black; |
| **Class:** Shiny cowbird | **Class:** Shiny cowbird    **Class:** Fish crow | **Shared Part:**<br>• Bill shape is all purpose;<br>• Bill length is shorter than head;<br>… |
| **Class:** Herring gull | **Class:** Western gull    **Class:** Glaucous gull | **Shared Part:**<br>• Bill shape is hooked;<br>• Bill length is the same as head;<br>… |
| **Class:** Herring gull | **Class:** Glaucous gull    **Class:** California gull | **Shared Part:**<br>• Belly color is white;<br>• Belly pattern is solid;<br>… |
| **Class:** Caspian tern | **Class:** Common tern    **Class:** Caspian tern | **Shared Part:**<br>• Wing color is white;<br>• Wing shape is long;<br>• Wing pattern is solid; |
| **Class:** Caspian tern | **Class:** Caspian tern    **Class:** Elegant tern | **Shared Part:**<br>• Tail shape is forked;<br>• Tail color is white;<br>• Tail pattern is solid; |

Figure 2: Success case of deliberative visualizations for images from CUB200 [7].