[Reviews · NeurIPS 2019]

Reviewer 1



The task is novel and methodology uses existing work on computing attribution maps using Taylor series approximation to compute ambiguity maps. The use and analysis of the effects of hardness scores are interesting. The work has a clear motivation (I like Fig 1), uses sound methodology, and relevant empirical experiments and so it meets the requirements for a NeurIPS-like conference. Overall, I find the paper clear enough to understand and the figures useful, especially the landing figure and the output figures. Although, I think are many easy fixes that can make the paper much more readable. Here are some suggestions (in chronological order): p. Choose a different visual illusion or remove it: I like the idea of using a visual illusion because it explains the task. But the current illusion it is not very convincing because--to me-- the individual image regions are not ambiguous but the entire image is. The bird example actually is pretty convincing and real. q. Line 72: "This observation is quantified by the introduction of a procedure to measure the alignment between network insecurities and attribute ambiguity, for datasets annotated with attributes." This is sentence is not clear, at all. r. From the introduction, it is not clear why we care about difficulty scores for generating deliberative explanations. s. Make the definition of deliberative explanation more visible: The most important part of the beginning of Section 3 is the definition of deliberative explanations, and in your current writeup, the definition is a tiny sentence at the end of second paragraph. Also, the first paragraph can be easily made more concise to make room for t. It is not clear what the green dots in the figures are and why they are only available for the bird classification dataset. u. a. Line 70: "For example, in Figure 1, insecurity 4 is caused by the presence of “black legs” ....” How do you know it is not caused by the presence of concept “two pointy sticks”?

Reviewer 2



I think that this is a good and well-written paper. Section 2 has a relevant review of related approaches. It is fairly easy to follow the arguments in most places of the paper and notation is neat; though, sections 3.1, 3.2, and 3.3 are quite compressed, and I could not understand the details deeply even though notation is very good in those sections. The examples in Fig. 1 (right) are very useful and encouraging, but the authors could clarify if those particular examples where generated by their methods. In Table 1, third section, the improvement of the second-order methods is rather small (small effect size). It "always outperforms" as the authors said, but I am not sure if this is a clear improvement. Do other results also confirm the improvements due to the second-order method?

Reviewer 3



Strength (beyond the points mentioned in the contribution): 1. This paper establishes a new framework for interpretability evaluation. The proposed method extends interpretability mining to the granularity level of attribution and enhances it with ambiguity class pairs. 2. This paper is well-organized. The author provides motivation with detailed examples and mathematical formulations to quantify the concept of insecurities. 3. This paper has a rigorous experimental design and convincing evaluation standard. The comparison on different formulation of difficulty score is particularly relevant to this field of research since they stand for different rationale for quantifying ambiguity. Weakness: A limitation of this method is that it needs to be trained on datasets that have strong annotation of feature. Most classification dataset lacks such annotation. A reasonable next step of research could focus on the automatic discovery and proposal of features that are relevant for the downstream tasks.

[Author Response · NeurIPS 2019]



Figure 1: MTurk interface

Figure 2: Robustness to shifts on CUB200.

| Score (attribution function) | $\rho$ | p-value |
|---|---|---|
| hesitancy score(gradient) | 0.58(0.06) | 5e-8(7e-8) |
| entropy score(gradient) | 0.38(0.11) | 8e-3(1e-2) |
| hardness score(gradient) | 0.65(0.05) | 7e-12(6e-12) |
| hardness score(integrated gradient) | 0.65(0.07) | 6e-10(9e-10) |
| hardness score(gradient-Hessian) | 0.69(0.06) | 9e-12(7e-12) |

Table 1: Pearson correlation coefficient ($\rho$) and p-value (mean(stddev)) on CUB200.

We start by thanking all reviewers for the careful consideration of the paper and the many suggestions for improvement. All the excellent suggestions regarding writing or presentation will be implemented in the revised version. Below, we address the comments that motivated further experiments or, in our viewpoint, required further clarification.
**R1.p-u, R1.2, R1.6)** All these points will be corrected in the new version. We note that our intention was not to hide the limitations of the approach (see reply to **R4** for more details).

**R1.1)** The two approaches have different motivations. Contrastive explanations seek regions or language descriptions explaining why an image does not belong to a counter-class. Deliberative explanations seek insecurities, i.e. the regions that make it difficult for the model to reach its prediction. To produce *an explanation,* contrastive methods only need to consider two *pre-specified* classes (predicted and counter), deliberative explanations must consider *all* classes and determine the ambiguous pair *for each region.* Comparing specifically to the Hendricks paper, it extracts a set of noun phrases from the counter-class and filters them with an evidence checker. Since phrases are defined by attributes, this boils down to detecting presence/absence of attributes in the image. Attribute annotation is needed for training. Deliberative explanations require characterizing the uncertainty of classification in every image region. They do not require attribute annotations, which are only needed for performance evaluation.

**R1.3)** We performed a preliminary human evaluation on MTurk, using the interface of Figure 1. Given an insecurity $(\mathbf{r}, a, b)$ found by the explanation algorithm, turkers were shown $\mathbf{r}$ and asked to identify $(a, b)$ among 5 classes (for which a random image was displayed): the two classes $a$ and $b$ found by the algorithm and 3 other random classes. Turkers agreed amongst themselves on $a$ and $b$ for $59.4\%$ of the insecurities and $33.7\%$ of randomly cropped regions. Turkers agreed with the algorithm for $51.9\%$ of the insecurities and $26.3\%$ of the random crops. This shows that 1) insecurities are much more predictive of the ambiguities sensed by humans, and 2) the algorithm predicts those ambiguities quite well. In both cases, the "Don't know" rate was around $12\%$.

**R1.4)** The importance of insecurity $(\mathbf{r}, a, b)$ was defined as $\frac{1}{|\mathbf{r}|}\sum_{i,j\in\mathbf{r}} m_{i,j}^{(a,b)}$. To determine how insecurities contribute to prediction, we measured the Pearson correlation coefficient $\rho$ between this score and insecurity precision defined in Section 4. Table 1 shows a strong positive correlation for non-self-referential scores and a moderate one for self-referential ones.

**R1.5)** Test images were randomly translated by 1 to 5 pixels and insecurities compared to those without translation. The similarity between two insecurities of ambiguities $(a, b)$ was then measured by the IoU metric, $\frac{|\{i|\mathbf{p}_i\in\mathbf{r},a_i=a,b_i=b\}\cap\{i|\mathbf{p}_i\in\mathbf{r}',a_i=a,b_i=b\}|}{|\{i|\mathbf{p}_i\in\mathbf{r},a_i=a,b_i=b\}\cup\{i|\mathbf{p}_i\in\mathbf{r}',a_i=a,b_i=b\}|}$, where $\mathbf{p}_i$ is as defined in Section 4. The average IoU across all ambiguities and examples is shown in Figure 2 as a function of the threshold $T$ of L172. The average IoU was almost always above $80\%$, which is a fairly high value. This suggests that insecurities are quite robust to image shifts.

**R2)** Fig. 1: insecurities are generated by our method. Tab 1, section 3: Fig. 2 (right) confirms the improvements of second-order attributions. [1] (published after NeurIPS deadline, will be cited) found experimentally that gains of second-order term decrease as the number of classes increases. This could explain why gains on ADE20K (>1000 categories) are smaller than on CUB200 (200 categories).

**R4)** Note that we do not need strong annotations for training, only class labels. The strong annotations (parts and attributes on CUB200, segmentation masks on ADE20K) are just needed to evaluate explanation performance, i.e. on the test set (see, e.g., footnote 1). Hence, the requirement for annotations is a limitation but only for the *evaluation* of deliberative explanation methods, not for their *use* by practitioners. Furthermore, it enables *reproducible* evaluation, which is not always the case for explanation methods.

[1] Singla, Sahil, et al. Understanding Impacts of High-Order Loss Approximations and Features in Deep Learning Interpretation. ICML, 2019.

[Meta-Review · NeurIPS 2019]

The reviewers agree that the paper should be accepted and I think this is a correct assessment. The idea proposed in the paper is novel, sound and well executed. The paper is also on a topic which is of interest to many participants of the conference.